# SHAP-PGD: A Realistic Adversarial Attack on Tabular Data by Unifying Interpretability and Semantic Consistency

## Abstract

Adversarial attacks on tabular data have unique challenges due to inter-feature constraints and semantic realism requirements, compelling attackers to introduce minimal perturbations to as few features as possible to generate realistic adversarial samples. However, existing methods overly relax or even overlook these constraints and become trapped in local optima. Furthermore, current research often neglects to interpret how adversarial samples perturb model decisions since the highly abstract nature of tabular data, with their semantic consistency evaluation remaining ambiguous and poorly defined. To address these challenges, we propose SHAP-PGD, an interpretable white-box tabular adversarial attack under complex constraints. SHAP-PGD utilizes global attribution to identify the most influential features and uses a decoupled gradient masking mechanism within this selected constraint-satisfying subspace to avoid local minima. This design enables the generation of realistic perturbations and enhances interpretability throughout the attack process. In addition, to investigate semantic consistency, we draw on both synthetic distribution and model utility, providing a concrete and scalable formulation of the consistency problem. Extensive experiments on four datasets and five victim architectures show that SHAP-PGD maintains semantic consistency while generating realistic perturbations and outperforms existing methods in 35 out of 40 settings with an average robustness reduction of 59.4%.

## 1 Introduction

Adversarial attack is the process of introducing semantically-preserving perturbation into input data to cause a model to produce incorrect answers. Although attack methods have been widely studied in computer vision and natural language processing (Ma et al., 2021; Wang et al., 2023), their adaptation and application on tabular data, the predominant data modality in various domains concerning human welfare (e.g., finance and healthcare), is still underexplored. Meanwhile, recent studies on tabular data prediction exhibit unique significance and usability of deep learning models compared to traditional tree-based models (Borisov et al., 2022; Yan et al., 2023; Chen et al., 2024), making it an increasingly urgent demand to establish corresponding assessment on their robustness. However, attacking tabular data is challenging: heterogeneous features (attributes) are often constrained by real-world rules and encode sensitive and interdependent information (e.g., income, age, or logical relations like birth date before death). Therefore, to avoid disrupting logical relations, attackers are required to adjust as few modifiable features as possible with minimal value changes.

Existing attack methods for tabular data either (i) apply rule-based feature selection, where attackers utilize domain knowledge to manually identify a set of modifiable features that respect inherent constraints (Kireev et al., 2022; Kulynych et al., 2018), or (ii) incorporate domain constraints explicitly into the optimization objective, such as employing projected gradient descent (Simonetto et al., 2024a; 2021). However, these methods neglect to control variations in both feature count and values, which are usually measured by $L_2$ or $L_\infty$ norms, overlooking practical scenarios where attackers are only required to adjust a sparse feature subset ($L_0$) with minimal perturbation ($L_{\{2,\infty\}}$) (Ben-Tov et al., 2024). Such variation requirements result in a highly non-convex and intractable search space, and few attempts were explored but all failed to strictly satisfy the combined constraints (Sheatsley et al., 2021; 2020; Bostani et al., 2022) or got trapped in local minima (Ben-Tov et al., 2024; Mathov et al.,

2022), leading to inefficient and suboptimal attacks. Besides, due to the highly abstract nature of tabular data, the semantic consistency and interpretability are often implicitly acknowledged but rarely formalized as an effectiveness evaluation perspective (Kireev et al., 2022; Mathov et al., 2022; Sheatsley et al., 2021). There are a few studies considering such feature semantic preservation (Ballet et al., 2019; Cartella et al., 2021), while their evaluations rely on prior domain-expert knowledge to define feature importance. In summary, adversarial attack on tabular data faces two core challenges:

***C1: Insufficient realistic perturbations under complex constraints.*** Real-world tabular data follows underlying inter-feature constraints. To ensure the realism, perturbations must be sparse ($L_0$) and minimal ($L_{\{2,\infty\}}$), leading to a highly non-convex space that makes optimization exceedingly difficult.

***C2: Insufficient considerations on semantic consistency and interpretability.*** Due to the highly abstract nature of tabular data, current studies on tabular attack blindly emulate attack paradigm from vision or language tasks generally assuming no affect on original data semantics, which underscore the need for semantic consistency evaluation and interpretability of how attacks affect model decisions.

To tackle these challenges, we propose **SHAP-guided Projected Gradient Descent (SHAP-PGD)**, an interpretable white-box attack method that synergistically integrates global feature attribution from SHAP (Shapley Additive Explanations) with local gradient optimization. SHAP-PGD operates in two phases: (**1**) *Global shapley guidance*, SHAP identifies the Top-$K$ features most responsible for misclassification, filtered by practical constraints to ensure realistic perturbations; (**2**) *Local gradient optimization*, gradient ascent is performed within this selected constraint-satisfying subspace, using a decoupled gradient masking mechanism. In forward propagation, perturbations are restricted to Top-$K$ feasible features, while backward propagation leverages gradients from all feature dimensions weighted by SHAP, maintaining global awareness and avoiding premature reduction of the search space. This two-phase design enables generation of realistic adversarial samples and enhances interpretability throughout the attack process. Finally, to quantitatively assess semantic consistency, we evaluate SHAP-PGD's adversarial samples using both synthetic distribution metrics and downstream model utility, providing a concrete and scalable formulation of the consistency problem.

Our **main contributions** are: (**i**) We propose SHAP-PGD, an interpretable white-box attack that integrates feature attribution with gradient optimization, enabling the generation of realistic perturbations under constraints with improved interpretability. (**ii**) We introduce a decoupled gradient masking mechanism that maintains a focused search in the forward pass while leveraging global feature attribution information in the backward pass, significantly improving attack efficacy and mitigating local optima. (**iii**) We provide a assessment of semantic consistency on tabular data, through both synthetic distribution and model utility, offering a concrete formulation for evaluating attack quality. (**iv**) Extensive experiments on four real-world datasets with five victim architectures demonstrate the effectiveness and semantic consistency of SHAP-PGD. We also innovatively use synthetic constraints to simulate physical environments, demonstrating that SHAP-PGD can maintain its performance under extreme conditions and showcasing its scalability.

## 2 PRELIMINARIES

We focus on the problem of realistic adversarial attacks on tabular data under complex constraints. Let $x \in \mathbb{R}^d$ denote the input sample, $y \in \{1, \ldots, C\}$ its corresponding label, and $h : \mathbb{R}^d \to \mathbb{R}^C$ the classifier, where $h_c$ is the output score for class $c$. Let $\Delta \subseteq \mathbb{R}^d$ denote the space of permissible perturbations. The adversarial objective is to find $\delta \in \Delta$ such that

$$\arg \max_{c \in \{1, \ldots, C\}} h_c(x + \delta) \neq y. \tag{1}$$

In contrast to image-based attacks, which typically require only an $L_p$ bound on the perturbation (i.e., $\Delta_p = \{\delta \in \mathbb{R}^d \mid \|\delta\|_p \leq \epsilon\}$ for $p \in \{2, \infty\}$), attacks on tabular data must further satisfy complex, domain-specific feature constraints due to the real-world meaning of each feature.

**Adversarial Attacks in the Tabular Domain.** Let $\phi : Z \to \mathbb{R}^d$ denote a feature mapping from object space $Z$ to a $d$-dimensional feature space $F = \{f_1, f_2, \ldots, f_d\}$. Each object $z \in Z$ is associated with a set of natural constraints. In the feature space, these constraints are represented by a set $\Omega$. For input $x$ obtained from $z$, we write $x \models \Omega$ if $x$ satisfies $\Omega$. Hence, a valid adversarial example for tabular data must both induce misclassification and satisfy $x + \delta \models \Omega$.

**Complex Constraints $\Omega$.** Following prior work (Simonetto et al., 2021; 2024a;b), we consider four constraint classes covering all typical constraints in empirical tabular benchmarks: (i) **immutability**,

specifying attacker-invariant features; (ii) **boundaries**, restricting features within predefined ranges; (iii) **type**, enforcing data types (continuous, discrete, categorical); and (iv) **feature relationships**, capturing dependencies among features. Formally, feature relationship constraints are defined by:

$$\omega ::= \omega_1 \wedge \omega_2 \mid \omega_1 \vee \omega_2 \mid \psi_1 \bowtie \psi_2, \quad \psi ::= c \mid f_i \mid x_i \mid \psi_1 \oplus \psi_2, \tag{2}$$

where $\bowtie \in \{<, \leq, =, \neq, \geq, >\}$ represents comparison operators, $\oplus \in \{+, -, \times, \div\}$ arithmetic operators, $c$ a constant, $f_i$ the current feature value, and $x_i$ its original value. Consider a real-world example, where the loan term can only be either 36 months or 60 months, and the number of open accounts must not exceed the maximum allowed for each customer. It can be formally expressed as:

$$\omega_1 = ((f_{\text{term}} = 36) \vee (f_{\text{term}} = 60)) \wedge (f_{\text{open\_acc}} \leq f_{\text{total\_acc}}) \tag{3}$$

**Realistic Adversarial Attacks in the Tabular Domain.** Following recent work and practical settings (Sheatsley et al., 2021; 2020; Bostani et al., 2022), we consider a realistic attack scenario, where only a small number of features are perturbed with limited magnitude to minimize attack cost. Specifically, let $\delta = s \odot m$ where $s \in \mathbb{R}^d$ is the perturbation magnitude vector and $m \in \{0, 1\}^d$ indicates the perturbed feature positions. A realistic adversarial perturbation on tabular data must satisfy:

$$(x + \delta) \models \Omega, \quad \|s\|_p \leq \epsilon, \quad (\|m\|_0 = \sum_{i=1}^{d} m_i) \leq k, \tag{4}$$

where $k$ denotes the maximum allowable number of perturbed features and $p \in \{2, \infty\}$. To enforce complex constraints, CPGD (Simonetto et al., 2021) transforms each constraint $\omega$ into a differentiable penalty function: $penalty(x, \omega)$, which evaluates to zero when $x$ satisfies $\omega$ and otherwise reflects the degree of violation. The attacker's objective thus becomes maximizing:

$$\max_{\|\delta\|_0 \leq k, \|\delta\|_p \leq \epsilon} \mathcal{L}(\theta, x + \delta) = \max_{\|\delta\|_0 \leq k, \|\delta\|_p \leq \epsilon} l(x + \delta, y) - \sum_{\omega_i \in \Omega} \text{penalty}(x + \delta, \omega_i), \tag{5}$$

where $\theta$ denotes the model parameters, $l$ is the loss function (typically the cross-entropy loss).

# 3 METHODS

In this section, we propose SHAP-PGD, an interpretable white-box attack method that enforces complex constraints using penalty functions and employs a two-phase optimization strategy: a ***global shapley guidance phase*** and a ***local gradient optimization phase*** to generate realistic perturbations. To comprehensively evaluate attack quality, beyond existing evaluation protocols, we provide a quantitative assessment of semantic consistency through both synthetic distributions and model utility, offering a concrete formulation for evaluating the effectiveness of adversarial attacks.

## 3.1 SHAP-GUIDED PROJECTED GRADIENT DESCENT

**Design Motivation.** Previous methods typically either generate a global perturbation under an $L_p$ norm and then select $k$ features to modify or simultaneously generate and select features within the optimization objective (Ben-Tov et al., 2024; Bostani et al., 2022; Kireev et al., 2022). These methods often lead to local optima and make it difficult to strictly enforce the joint $L_0 \& L_p$ norm. Moreover, only a few works leverage domain-specific feature importance to interpret attacks, but such strategies require extensive manual effort and do not generalize well across datasets (Ballet et al., 2019; Cartella et al., 2021). To overcome these issues, SHAP-PGD unifies SHAP's global feature attributions with the efficient local search of PGD, allowing the generation of realistic perturbations that satisfy norms and avoid local optima. SHAP-PGD also provides interpretability throughout the attack process, without requiring any prior knowledge or manual intervention.

**Global Shapley Guidance Phase.** SHAP is an interpretability method based on Shapley values from cooperative game theory (Lundberg & Lee, 2017), widely used in structured data modeling (Ribeiro et al., 2016; Lundberg et al., 2018). The key idea is to attribute the model output to individual feature contributions. For the $j$-th feature of a sample $x$, its exact Shapley value is computed as:

$$\phi_j(x) = \sum_{S \subseteq F \setminus \{j\}} \frac{|S|!(d - |S| - 1)!}{d!} \left[ h_y\left(\mathbf{x}_{S \cup \{j\}}\right) - h_y\left(\mathbf{x}_S\right) \right], \qquad j = 1, \ldots, d, \tag{6}$$

where $\mathbf{x}_S$ with only feature subset $S$ present. This measures the marginal impact of feature $j$ on the output. To rank features independent of sign, we use the normalized absolute Shapley value:

$$\hat{g}_j(x) = \frac{|\phi_j(x)|}{\max_{\ell \in F} |\phi_\ell(x)| + \varepsilon_{\text{num}}}, \qquad \hat{g}_j(x) \in [0,1], \tag{7}$$

with $\varepsilon_{\text{num}} = 10^{-7}$ to avoid division by zero. SHAP importance provides more robust and interpretable guidance than gradient-based heuristics. However, computing Eq. 6 exactly is intractable for large $d$ due to exponential complexity. **In the white-box setting**, we approximate Shapley values using a gradient-based approach inspired by Integrated Gradients (IG) (Sundararajan et al., 2017):

$$\phi_j(x) = \frac{1}{|N|} \sum_{n=1}^{|N|} \left( x_j - z_j^{(n)} \right) \underbrace{\int_0^1 \frac{\partial h_y\left( z^{(n)} + \eta(x - z^{(n)}) \right)}{\partial x_j} \, d\eta}_{\text{IG path integral}}, \tag{8}$$

where $N = \{z^{(1)}, \ldots, z^{(|N|)}\}$ are training samples and $\eta \in [0,1]$. This averages integrated gradients along straight-line paths from each $z^{(n)}$ to $x$, weighted by feature difference, efficiently approximating global feature attribution. This reduces complexity from $\mathcal{O}(2^d)$ to $\mathcal{O}(|N| d)$. The top-K features with highest importance are then selected to form the attack mask, ensuring $L_0$ sparsity:

$$m_j = \begin{cases} 1, & j \in \text{TopK}\left( \hat{g}_j(x)_{j=1}^d, k \right), \\ 0, & \text{otherwise.} \end{cases} \qquad m = [m_1, \ldots, m_d]. \tag{9}$$

**Local Gradient Optimization Phase.** We propose a joint optimization that combines globally computed Shapley values with gradient projection. This **decoupled gradient masking mechanism** enables realistic perturbations under both $L_0$ and $L_p$ constraints. We initialize the perturbation $s^{(0)} = 0$. For $T$ iterations, we compute a soft mask $m^{(t)} = \sigma(\hat{g}(x))$, where $\sigma(x) = \frac{1}{1+e^{-x}}$ is a sigmoid function, and construct the candidate adversarial example and loss as:

$$x_{\text{adv}}^{(t)} = \text{clip}_{[0,1]} \left( x + s^{(t)} \odot \Pi_0 m^{(t)} \right), \quad \mathcal{L}^{(t)} = \mathcal{L}\left( \theta, x_{\text{adv}}^{(t)} \right), \tag{10}$$

where $\Pi_0$ is a projection operator that enforces the $L_0$ constraint, $\odot$ is an element-wise product, and $\text{clip}_{[0,1]}(\cdot)$ ensures the output stays within the valid input range. The perturbation is updated by:

$$\widetilde{s}^{(t+1)} = s^{(t)} + \alpha \, \text{sign}\left( \nabla_{s^{(t)}} \mathcal{L}^{(t)} \right), \quad s^{(t+1)} = \Pi_{[-\epsilon,\epsilon]}\left( \widetilde{s}^{(t+1)} \right), \tag{11}$$

where $\alpha$ is the step size and $\nabla_{s^{(t)}} \mathcal{L}^{(t)} = \nabla_\delta \mathcal{L}^{(t)} \odot m^{(t)}$. The projection operator $\Pi_{[-\epsilon,\epsilon]}$ that enforces the $L_p$ constraint. No projection is applied to $m^{(t)}$ here, to avoid restricting the search space too early and thus falling into local optima. Unlike previous methods (Kireev et al., 2022; Cartella et al., 2021; Ben-Tov et al., 2024), we do not simply take $\delta^{(t+1)}$ as final, but update the latent mask:

$$m^{(t+1)} = m^{(t)} + \beta \frac{\nabla_{m^{(t)}} \mathcal{L}^{(t)}}{||\nabla_{m^{(t)}} \mathcal{L}^{(t)}||_2 + \varepsilon_{\text{num}}}, \tag{12}$$

where $\beta$ is the step size. This approach dynamically optimizes both the magnitude and position of perturbations, allowing less important features to be perturbed if advantageous. However, $m^{(t+1)}$ depends on $s^{(t)}$ (since $\frac{\partial \mathcal{L}}{\partial m^{(t)}} = \nabla_\delta \mathcal{L}^{(t)} \odot s^{(t)}$), and $\widetilde{s}^{(t+1)}$ depends on $m^{(t)}$, causing mutual dependence. If one variable is trapped in a local optimum, the other is likely to follow. To mitigate this, we use two strategies: **(1) Stagnation Detection:** If the Top-K features do not change over several iterations and the attack fails, we reinitialize their values (randomly or using updated SHAP scores). **(2) Periodic Correction:** Every $T_s$ steps, we recompute SHAP attributions and update $m^{(t+1)}$ as:

$$m^{(t+1)} = \begin{cases} \hat{g}(x_{\text{adv}}^{(t+1)}), & \text{if } (t+1) \bmod T_s = 0 \\ m^{(t+1)}, & \text{otherwise.} \end{cases} \tag{13}$$

These mechanisms jointly optimize perturbation magnitude and location, improving attack feasibility. The overall algorithm and corresponding complexity analysis are detailed in **Appendix A.1** and **A.2**.

**Theoretical Guarantees.** We provide theory to show that SHAP-PGD can generate realistic adversarial perturbations while satisfying complex constraints. See the **AppendixA.3** for details.

(a) Original correlation (b) SHAP-PGD correlation

(c) CAPGD correlation (d) MOEVA correlation

Figure 1: Pearson correlation heatmaps for original and adversarial samples on the URL dataset

Table 1: Evaluation of data distribution differences between original and adversarial samples generated by SHAP-PGD, CAPGD

| Methods | Dataset | Model | JSD | $d_{Wass.}$ | $\beta$-Score | $C$-Score |
|---|---|---|---|---|---|---|
| SHAP-PGD | LCLD | TabTr. | 0.037 | 0.018 | 0.912 | 0.816 |
| | | STG | 0.036 | 0.108 | 0.772 | 0.791 |
| | | TabNet | 0.054 | 0.134 | 0.862 | 0.772 |
| | WIDS | TabTr. | 0.102 | 0.092 | 0.831 | 0.874 |
| | | STG | 0.091 | 0.076 | 0.807 | 0.865 |
| | | TabNet | 0.053 | 0.079 | 0.948 | 0.899 |
| CAPGD | LCLD | TabTr. | 0.348 | 0.330 | 0.461 | 0.438 |
| | | STG | 0.149 | 0.359 | 0.353 | 0.552 |
| | | TabNet | 0.183 | 0.202 | 0.515 | 0.504 |
| | WIDS | TabTr. | 0.158 | 0.237 | 0.367 | 0.460 |
| | | STG | 0.138 | 0.156 | 0.510 | 0.530 |
| | | TabNet | 0.370 | 0.198 | 0.635 | 0.554 |

Table 2: Evaluation of clean test dataset performance, where the model is fitted on the clean and attacked training datasets respectively

| Dataset | Model | Clean | CAPGD | Sarse-PGD | CaFA | SHAP-PGD |
|---|---|---|---|---|---|---|
| URL | TabTr. | 93.6 | 81.3 | 86.4 | 91.2 | **94.9** |
| | LightGBM | 97.1 | 93.7 | 95.3 | 96.7 | **98.5** |
| LCLD | TabTr. | 69.5 | 64.5 | 61.3 | 68.2 | **71.0** |
| | LightGBM | 70.6 | 68.8 | 64.5 | 70.0 | **72.1** |
| WIDS | TabTr. | 75.5 | 70.2 | 72.1 | 75.0 | **77.3** |
| | LightGBM | 82.0 | 81.1 | 77.0 | 81.6 | **83.5** |

### 3.2 SEMANTIC CONSISTENCY ASSESSMENT

**Design Motivation.** Due to the high abstraction inherent in tabular data, semantic consistency assessment is often overlooked in existing research. Only a few methods address this issue, but they tend to provide only implicit or informal definitions of consistency (Kireev et al., 2022; Mathov et al., 2022; Sheatsley et al., 2021). To comprehensively investigate semantic consistency, we draw on both synthetic distribution and model utility, proposing two hypotheses: **(1) The closer adversarial examples are to real data, the more likely they are to maintain the original semantics; (2) If a tabular model trained on adversarial data achieves comparable performance on the test set, it indicates that the adversarial samples preserve the original semantics.**

**For hypothesis 1,** we compare the distributions of original and adversarial samples generated by SHAP-PGD, CAPGD and MOEVA. As shown in Fig. 1, Pearson correlation heatmaps for the URL dataset indicate that SHAP-PGD's adversarial samples closely align with the original distribution, while CAPGD and MOEVA introduce larger shifts, suggesting that enforcing joint $L_0 \& L_p$ feasibility better preserves real-world feature dependencies. We further use four standard synthetic data metrics (Qian et al., 2023)—Jensen-Shannon Divergence (JSD), Wasserstein Distance ($d_{Wass.}$; both lower is better), and $\beta$-Score and $C$-Score (both higher is better). As show in Table 1, the adversarial samples generated by SHAP-PGD achieve an average JSD of 0.063, compared to 0.227 for CAPGD, with similarly superior performance on the other metric. These results demonstrate that SHAP-PGD's adversarial samples are nearly indistinguishable from real data across multiple datasets.

**For hypothesis 2,** inspired by synthetic images using models to assess data quality (Heusel et al., 2017), we separately train models on the clean training set and on adversarial samples generated by various attack methods, then evaluate accuracy on the clean test set. As shown in Table 2, across all datasets, models trained on SHAP-PGD adversarial samples outperform those trained on other types of adversarial data, with an average improvement of 1.6% compared to models trained on clean data. In contrast, models retrained with CAPGD and Sparse-PGD samples suffer average performance drops of 4.6% and 6.3%, respectively. These results indicate that SHAP-PGD generates more realistic adversarial samples, providing empirical support for their semantic consistency with the original data.

## 4 EXPERIMENTS

In this section, we conduct experiments on four real-world datasets to evaluate the performance of SHAP-PGD, focusing on the following questions:

- **Q1:** How does SHAP-PGD perform compared to state-of-the-art attack methods?
- **Q2:** In what ways do perturbations introduced by SHAP-PGD affect the model's decision process?
- **Q3:** How do perturbation budgets and complex constraints influence the SHAP-PGD?

Table 3: Evaluation on robust accuracy. The Clean column reports accuracy on unperturbed samples; lower robust accuracy indicates a more effective attack. We highlight the lowest robust accuracy in **bold**. Following prior work, we set $L_2 = 0.5$ and $L_0 = 0.1$, i.e., after feature normalization, we perturb 10% of the features with a magnitude of 50%

| Dataset | Model | Clean | $L_2 = 0.5$ | | | | $L_2 = 0.5 \& L_0 = 0.1$ | | | |
|---|---|---|---|---|---|---|---|---|---|---|
| | | | CAPGD | BF* | MOEVA | SHAP-PGD | PGD$_0$ | Sparse-PGD | CaFA | SHAP-PGD |
| URL | TabTr. | 93.6 | 10.9±0.1 | 93.2±0 | 18.2±0.8 | **2.1±0.3** | 31.2±0.5 | 6.4±0.5 | 25.2±0.7 | **3.7±1.0** |
| | RLN | 94.4 | 12.6±0.2 | 93.8±0 | 23.6±0.5 | **7.5±0.5** | 25.5±0.8 | 22.3±0.6 | 27.8±0.7 | **10.4±0.5** |
| | VIME | 92.5 | 56.3±0.1 | 92.2±0 | 56.5±0.9 | **30.8±0.6** | 50.0±1.4 | 39.3±1.1 | 41.7±0.3 | **22.2±0.3** |
| | STG | 93.3 | 72.6±0.0 | 93.2±0 | 58.2±0.9 | **17.2±0.4** | 33.3±1.6 | 19.6±0.6 | 28.6±0.1 | **17.6±0.8** |
| | TabNet | 93.4 | 19.3±0.6 | 90.9±0 | 17.5±0.6 | **10.9±0.1** | 49.9±0.3 | 10.9±0.3 | 43.4±1.1 | **8.9±0.2** |
| LCLD | TabTr. | 69.5 | 27.1±0.9 | 61.1±0 | 10.7±0.8 | **5.4±0.1** | 20.7±2.0 | **9.1±0.2** | 12.5±0.2 | 14.3±0.6 |
| | RLN | 68.3 | 0.2±0.1 | 38.9±0 | 0.8±0.2 | **0.1±0** | 9.5±0.8 | 11.4±0.6 | 9.0±1.8 | **4.1±0.7** |
| | VIME | 67.0 | **2.6±0.2** | 52.6±0 | 24.1±1.5 | 4.9±0.3 | 30.1±0.2 | 31.0±0.4 | 29.4±0.7 | **17.2±0.6** |
| | STG | 66.4 | 55.5±0.2 | 53.0±0 | 55.4±0.2 | **50.2±0.3** | 62.3±0.1 | 59.7±0.7 | 60.1±0.9 | **57.7±0.6** |
| | TabNet | 67.4 | 6.3±0.4 | 49±0 | **0.8±0.1** | 9.4±0.1 | 55.8±0.8 | 20.2±0.6 | 38.0±1.2 | **15.1±0.8** |
| CTU | TabTr. | 95.3 | 95.3±0 | 95.3±0 | 95.3±0 | 95.3±0 | 95.3±0 | 95.3±0.0 | 95.3±0 | 95.3±0 |
| | RLN | 97.8 | 97.8±0 | 97.5±0 | 94.0±0.2 | **88.9±0.3** | 94.8±0.2 | 96.2±0.3 | 95.0±0.7 | **91.1±0.3** |
| | VIME | 95.1 | 95.1±0 | 95.1±0 | 40.8±4.7 | **38.4±0.9** | 68.5±0.1 | 44.5±0.2 | 47.9±1.4 | **22.6±1.0** |
| | STG | 95.3 | 95.3±0 | 95.3±0 | 95.3±0 | 95.3±0 | 95.3±0 | 95.3±0 | 95.3±0 | 95.3±0 |
| | TabNet | 96.1 | 96.1±0 | 13±0 | **0.0±0** | **0.0±0** | 12.3±0.6 | 4.6±1.8 | 7.3±0.5 | **3.4±0** |
| WIDS | TabTr. | 75.5 | 48.0±0.3 | 67.7±0 | 59.2±0.6 | **45.4±0.5** | 63.6±0.5 | 65.9±0.6 | 61.3±0.2 | **51.3±0.8** |
| | RLN | 77.5 | 61.8±0.3 | 77.0±0 | 67.7±0.3 | **53.5±0.3** | 76.3±0.2 | 73.3±0.5 | 65.1±0.3 | **59.5±0.4** |
| | VIME | 72.3 | 51.4±0.3 | 71.2±0 | 59.4±0.5 | **42.0±0.9** | 66.8±0.5 | 61.8±1.1 | 62.1±1.3 | **50.8±0.7** |
| | STG | 77.6 | 65.1±0.4 | 77.5±0 | 68.8±0.3 | **61.6±0.1** | 70.0±0.2 | 67.9±0.3 | **63.0±2.3** | 69.1±1.2 |
| | TabNet | 79.7 | 10.2±0.3 | 73.1±0 | 13.9±0.4 | **1.6±0.3** | 21.5±1.3 | **2.3±0.5** | 10.8±0.6 | 5.1±0.8 |

- **Q4:** What is the computational efficiency of SHAP-PGD?
- **Q5:** How do hyperparameters and individual components contribute to the SHAP-PGD?
- **Q6:** How does SHAP-PGD perform in physical environments, especially in extreme situations?
- **Q7:** How is SHAP-PGD scalable, especially in terms of transferability and adversarial training?

## 4.1 EXPERIMENTAL SETTINGS

**Dataset.** Following previous works (Simonetto et al., 2024a;c;b), We evaluate on four public tabular datasets from diverse domains. URL (Hannousse & Yahiouche, 2021), LCLD (George, 2018), WiDS (Lee et al., 2020) and CTU (Chernikova & Oprea, 2022).

**Architectures.** Following Simonetto et al. (2024a), we test five widely used architectures: TabTransformer (TabTr.) (Huang et al., 2020), TabNet (Arik & Pfister, 2021), RLN (Shavitt & Segal, 2018), STG (Yamada et al., 2020) and VIME (Yoon et al., 2020).

**Baselines.** We compare SHAP-PGD against sparse, norm-constrained attacks enforcing $L_0$ and $L_{\{2,\infty\}}$ feasibility. Gradient-based baselines include PGD$_0$ (Croce & Hein, 2019), Sparse-PGD (Zhong et al., 2024), CaFA (Ben-Tov et al., 2024), and CAPGD (Simonetto et al., 2024c). We also evaluate publicly available attacks for continuous features, BF* (Kulynych et al., 2018) and MOEVA (Simonetto et al., 2021). Some methods were originally reported only under $L_0$ or only under $L_{\{2,\infty\}}$; where applicable, we implement the complementary norm and evaluate those methods under both. We exclude ensemble attacks (CAA (Simonetto et al., 2024a), sAA (Zhong et al., 2024)) because their constituent attacks are already included.

**Evaluation metrics.** Attack effectiveness is measured by robust accuracy, i.e., classification accuracy on valid adversarial samples. Unchanged misclassified samples remain unperturbed, and invalid adversarial examples are counted as correct.

**Supplement.** All datasets, architectures, baselines and implementation details are in **Appendix A.4**.

## 4.2 OVERALL PERFORMANCE (*Q1*)

We first evaluate the robust accuracy of SHAP-PGD in comparison with other methods. As shown in Table 3, SHAP-PGD achieves the lowest robust accuracy in 18 out of 20 cases under the $L_2$ constraint, and in 17 out of 20 cases under the joint $L_2 \& L_0$ constraint, across various datasets and models. For instance, on the URL dataset, SHAP-PGD reduces the robust accuracy of TabTransformer from 93.6% (clean) to just 2.1% ($L_2$) and 3.7% ($L_2 \& L_0$), significantly outperforming all competing methods—a trend that is consistent across other datasets. Moreover, SHAP-PGD demonstrates superior stability, with an average standard deviation of 0.48 and 0.74 for the $L_2$ and $L_2 \& L_0$ settings respectively, both notably lower than those of other state-of-the-art attack methods. These results collectively verify the effectiveness and reliability of SHAP-PGD as an adversarial attack tool.

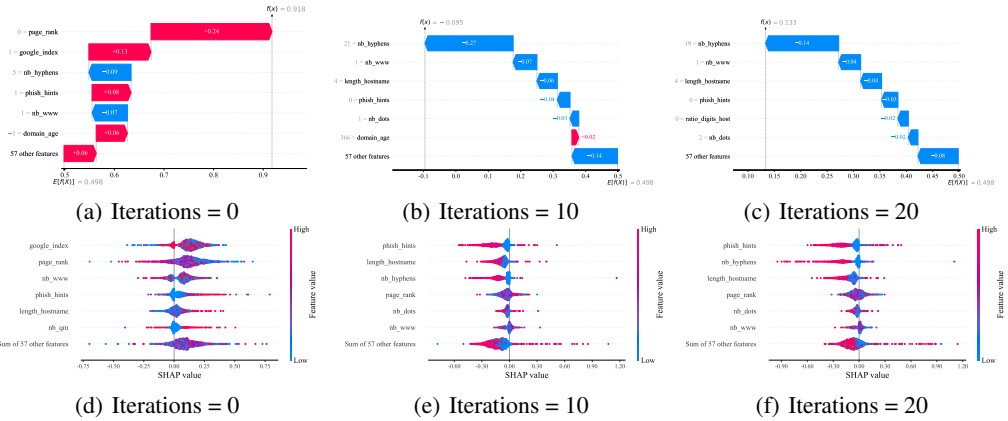

Figure 2: The impact of adding perturbations to URL dataset samples on model decisions

### 4.3 INTERPRETABLE PERTURBATION STUDY (Q2)

To study interpretability, SHAP-PGD leverage shapley values to show how each perturbation alters the model's decision process. Using the URL dataset and TabTransformer (with similar findings on other datasets), we analyze two perspectives:

- **Instance-level Analysis.** Waterfall plots (top row of Fig. 2) illustrate how perturbations influence SHAP attributions for individual samples. Each plot decomposes model output into feature-wise effects (red for positive, blue for negative, x-axis in log-odds and features are sorted by descending importance on the y-axis). For example, at iteration 0, class 1 prevails ($f(x) = 0.918$) due to strong positive contributions from *page_rank* and *google_index*, offsetting the negative impact of *nb_hyphens*. By iteration 10, the negative effect of *nb_hyphens* intensifies (from -0.09 to -0.27), flipping the prediction to class 0 ($f(x) = -0.095$). SHAP-PGD's feature selection frequency reveals the top three perturbed features: ***length_hostname (9,035), phish_hints (7,938), nb_hyphens (7,623)***, consistent with their ranked importance shown in the figure.
- **Population-level Analysis.** To assess the overall impact on the model, we visualize 2,000 class-1 samples using beeswarm plots (second rows of Fig. 2). Prior to the attack, feature contributions are concentrated on the positive side, while after 10 iterations, perturbations consistently shift feature attributions from positive to negative, causing values across different features to have adverse impacts on model decisions. By 20 iterations, this effect becomes even more pronounced.

### 4.4 IMPACT OF ATTACK BUDGET AND COMPLEX CONSTRAINTS (Q3)

We systematically evaluate the effects of attack budget and complex constraints on SHAP-PGD, focusing on: **(1) perturbation magnitude ($\epsilon$); (2) number of perturbed features ($k$); and (3) feasible success rate.** For the first two, we conduct experiments on the URL dataset (results on other datasets are in **Appendix A.5**). As shown in Fig. 3, robust accuracy declines sharply as $\epsilon$ increases, with a significant average drop of 54.4% between $\epsilon = 0.3$ and $\epsilon = 0.4$, after which performance plateaus. Solid and dashed lines correspond to $k = 0.1$ and $k = 0.2$, respectively. We further examine the effect of varying $k$ under fixed $\epsilon$ values ($\epsilon = 0.1$ for the solid line and $\epsilon = 0.2$ for the dashed line). Here, robust accuracy decreases approximately linearly with increasing $k$, showing a reduction of about 7.21% for every additional 10% of perturbed features. To evaluate adaptability under constraints, we define the feasible success rate: $\frac{|x_{adv}|x_{adv}|=\Omega \wedge x_{adv} \in X_{adv}|}{|X_{adv}|}$. Under $L_2 = 0.5$ and joint $L_0 = 0.1 \& L_2 = 0.5$, SHAP-PGD achieves an average feasible success rate of 88.46% across all architectures, surpassing all baselines (see Fig. 4). On the CTU dataset, the large number and complexity of constraints lead to low performance across all methods (see **Appendix A.6** for further analysis). Notably, SHAP-PGD's feasible success rate improves further under joint constraints ($L_0 \& L_2$), supporting our conclusion that the joint norm is more likely to maintain availability.

### 4.5 RUNTIME AND MODEL ABLATION STUDY (Q4 & Q5)

To answer Q4 and Q5, more experiments can refer to the **Appendix A.7** and **A.8**. Specifically, we evaluated several SHAP-PGD variants, including adaptive step size, momentum, and the difference of logits ratio loss function as proposed in Simonetto et al. (2024a); Croce & Hein (2020). While these techniques are effective under $L_2$ and $L_\infty$ in other settings, we found that they did not improve attack performance in the $L_0$ norm scenario and therefore were not adopted.

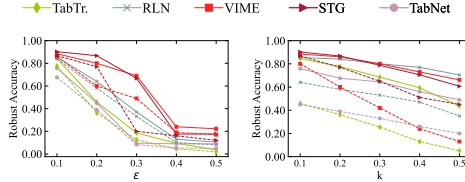 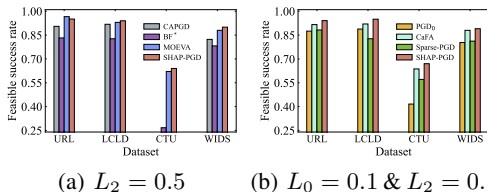

(a) $L_2 = 0.5$      (b) $L_0 = 0.1$ & $L_2 = 0.5$

Figure 3: The impact of perturbation budget on robust accuracy for the URL dataset

Figure 4: The feasible success rate of various baselines on four datasets

Table 4: Evaluation of robust accuracy on two physical simulation synthetic datasets

| Formulas | Constrains | Shapley Values | Model | Clean | CAPGD | CaFA | Sparse-PGD | SHAP-PGD |
|---|---|---|---|---|---|---|---|---|
| $y = \mathbb{I}(v_4 + \log \frac{v_3+2}{v_7+1} \leq 1)$ | $\omega_1 = ((v_1 = 0.3) \lor (v_5 = 0.6))$ $\omega_2 = (v_4 > v_8)$ $\omega_3 = ((v_3 + v_4) < \log v_9)$ | | TabTr. | 82.5 | 25.8 | 17.3 | 20.7 | **14.1** |
| | | | RLN | 81.1 | 43.9 | 36.8 | 40.4 | **25.2** |
| | | | VIME | 83.7 | 53.6 | **29.1** | 51.9 | 37.4 |
| | | | STG | 88.3 | 45.2 | 30.4 | 43.3 | **18.5** |
| | | | TabNet | 85.4 | 46.7 | 38.5 | 32.2 | **26.8** |
| $y = \mathbb{I}(2^{v_3} + \sin(v_1) + v_7 > 2)$ | $\omega_1 = ((v_2 = 0.1) \land (v_4 = 0.3))$ $\omega_2 = (v_3 - v_9) * v_5 < 0.1$ $\omega_3 = (\sin v_3 = 0.1)$ | | TabTr. | 59.8 | 48.2 | 29.8 | 32.4 | **30.1** |
| | | | RLN | 67.4 | 50.5 | 32.7 | 43.9 | **24.6** |
| | | | VIME | 66.9 | 51.2 | 43.6 | 44.8 | **31.9** |
| | | | STG | 60.4 | 48.3 | 31.1 | 32.5 | **29.7** |
| | | | TabNet | 70.2 | 53.4 | **31.2** | 35.9 | 36.6 |

## 4.6 PHYSICAL SIMULATION ATTACK PERFORMANCE STUDY (*Q6*)

To simulate the attack performance in a physical environment, we use synthetic constraints to evaluate under extreme conditions (e.g., immutable features have high shapley values, interdependent feature constraints). Each dataset consists of 80,000 training samples and 20,000 test samples, with 10 features independently generated from uniform noise in the range (0, 1] ($x = \{v_i | i = 0, \ldots, 9 \land v_i \in (0, 1]\}$). The formula in the first column of Table 4 is used to compute the classification label $y$, where $\mathbb{I}$ denotes the indicator function. To assess the performance under complex scenarios, we introduce a series of constraints for each dataset. In particular, we restrict the values of features with high shapley scores and impose correlations among features (in the third column of Table 4, darker colors represent higher values). Subsequently, we evaluate the robust accuracy on both the clean and adversarial test sets across five different architectures. CAPGD is constrained by $L_2 = 0.5$, while other methods are subject to $L_0 = 0.1$ & $L_2 = 0.5$. As shown in Table 4, SHAP-PGD outperforms other methods in 18 out of 20 settings. Compared to the clean baseline, SHAP-PGD yields an average robust accuracy drop of 47.1%, while CAPGD, CaFA, and Sparse-PGD exhibit declines of 28.0%, 43.5%, and 38.8% respectively. These results suggest that even when constraining features with high shapley values, SHAP-PGD is still able to maintain strong attack performance.

## 4.7 ATTACK SCALABLE STUDY (*Q7*)

In this section, we evaluate SHAP-PGD's scalable performance in ***transfer attacks*** and ***adversarial training***, under the joint constraint $L_0 = 0.1$ & $L_2 = 0.5$. We assess on TabTransformer, STG, and TabNet (with similar trends observed on other architectures).

**For transfer attacks,** we generate adversarial samples using each architecture and evaluate robust accuracy on other architectures. Table 8 (**in Appendix**) shows SHAP-PGD achieves high transferability compared to other methods. For example, on the URL dataset, using TabTransformer-generated samples on STG and TabNet, robust accuracy drops to 9.5% and 3.9%, respectively. We also test these examples on **tree-based models**. As show in Table 5, transfer attacks generated by SHAP-PGD reduce robust accuracy to averages of 41.3% for URL and 8.3% for LCLD. This shows that SHAP-PGD's attack has high transferability, whether on deep models or tree models.

**For adversarial training (AT),** we follow the framework by Madry et al. (2017), using PGD₀ ($L_0 = 0.1$ & $L_2 = 0.5$), recognized as a standard defense (Tramer et al., 2020). Our results highlight two points:***(1) realistic perturbations can improve clean performance.*** Unlike previous reports of degraded performance after AT (Simonetto et al., 2024a), we find a mean clean accuracy increase of 2.63%, with TabNet on WIDS improving by 16.8% (Table 6). ***(2) SHAP-PGD remains effective against with AT.*** Comparing Table 6 to Table 3, SHAP-PGD achieves an average improvement rate of 162% across four datasets, while Sparse-PGD and CaFA reach 434% and 283%. This confirms SHAP-PGD's effectiveness even against defended models.

Table 5: Robust evaluation of transfer attacks on tree-based models

| Dataset | Model | Clean | CaFA | SHAP-PGD |
|---|---|---|---|---|
| URL | Random Forest | 96.2 | 68.5 | **59.5** |
| | XGBoost | 97.4 | 43.6 | 35.2 |
| | LightGBM | 97.1 | 40.1 | 29.3 |
| LCLD | Random Forest | 64.3 | 31.7 | **9.5** |
| | XGBoost | 68.3 | 10.2 | **8.0** |
| | LightGBM | 70.6 | 18.9 | **7.5** |
| CTU | Random Forest | 95.6 | 95.6 | 95.6 |
| | XGBoost | 97.3 | 97.3 | **90.4** |
| | LightGBM | 96.9 | 96.5 | **87.7** |
| WIDS | Random Forest | 52.2 | 11.2 | **8.3** |
| | XGBoost | 80.4 | 51.1 | **42.2** |
| | LightGBM | 82.0 | 43.4 | **32.9** |

Table 6: Robust Evaluation in Adversarial Training between STG, TabNet and TabTr.

| Dataset | Model | Clean | Sparse-PGD | CaFA | SHAP-PGD |
|---|---|---|---|---|---|
| URL | $AT_{TabTr.}$ | 93.7 | 44.6 | 53.2 | **12.7** |
| | $AT_{STG}$ | 95.1 | 60.3 | 52.7 | **26.2** |
| | $AT_{TabNet}$ | 98.4 | 44.8 | 66.2 | **36.8** |
| LCLD | $AT_{TabTr.}$ | 74.3 | 35.1 | 41.8 | **32.9** |
| | $AT_{STG}$ | 68.0 | 63.4 | 65.3 | **60.4** |
| | $AT_{TabNet}$ | 70.5 | 50.7 | 47.9 | **30.9** |
| CTU | $AT_{TabTr.}$ | 95.3 | 95.3 | 95.3 | 95.3 |
| | $AT_{STG}$ | 95.3 | 95.3 | 95.3 | 95.3 |
| | $AT_{TabNet}$ | 95.6 | 39.5 | 40.1 | **37.2** |
| WIDS | $AT_{TabTr.}$ | 76.2 | 74.2 | **65.8** | 69.4 |
| | $AT_{STG}$ | 78.1 | 70.0 | **66.4** | 73.6 |
| | $AT_{TabNet}$ | 96.5 | 48.7 | 43.9 | **35.7** |

## 5 RELATED WORK

**Adversarial Attacks under Tabular Constraints.** Adversarial attacks on tabular data must strictly satisfy feature constraints. Existing methods are mainly white-box or black-box. White-box methods, such as LowProFool (Ballet et al., 2019), penalize key feature perturbations, with later works (Nobi & Krishnan, 2022; Teuffenbach et al., 2020; Alhussien et al., 2024) enforcing validity through type, value, and relational constraints, often via projection or differentiable penalties. Constrained I-FGSM variants (Tian et al., 2020; Kong & Ge, 2023) use binary masks and adaptive steps, while recent approaches employ SAT solvers or direct projection (Sheatsley et al., 2021; Ben-Tov et al., 2024; Simonetto et al., 2021; 2024a;c). Black-box methods use heuristics like feature replacement or sample synthesis (e.g., Flow-Merge (Abusnaina et al., 2019), FIGA (Gressel et al., 2021)) or metaheuristic multi-objective optimization (e.g., CoEVa2 (Ghamizi et al., 2020), MOEVA (Simonetto et al., 2021)). Existing methods either focus on $L_2$ or $L_\infty$ norms, neglecting real-world requirements for both sparsity ($L_0$) and minimal perturbation ($L_{\{2,\infty\}}$), or struggle to strictly constrain both $L_0$ and $L_p$, making them susceptible to local optima (Dyrmishi et al., 2025; Djilani et al., 2025). SHAP-PGD addresses these limitations by combining SHAP's global feature attribution with PGD's local search to jointly enforce $L_0$ and $L_{\{2,\infty\}}$. Our decoupled gradient masking allows dynamic exploration of the perturbation space, reducing local optima and generating more realistic adversarial examples.

**Semantic Consistency and Interpretability in Tabular Adversarial Attack.** Due to the highly abstract nature of tabular data, previous works preserve semantics by avoiding changes to class-relevant features (Chen et al., 2020; Debicha et al., 2023; Sun et al., 2023), while Mathov et al. (2022); Piatkowska & Smith (2020) enforce statistical metrics to maintain relevance to the target variable. Sheatsley et al. (2021) use category-specific constraints, but formulations of consistency remain implicit. Kireev et al. (2022); Sheatsley et al. (2020) even question the suitability of semantic consistency in tabular settings. For interpretability, Ballet et al. (2019) and Cartella et al. (2021) guide perturbations with feature importance, but require domain knowledge and manual effort. Shirazi et al. (2021; 2019) use pre-collected realistic feature values to replace the original features to ensure interpretability. In contrast, SHAP-PGD uses global feature attribution to explain how perturbations affect model decisions without prior knowledge. Additionally, we assess semantic consistency using synthetic distribution metrics and model utility, providing a concrete and scalable formulation.

## 6 CONCLUSIONS

This work tackles two challenges for attacks on tabular data: (**1**) generating realistic perturbations under complex constraints and (**2**) enabling comprehensive semantic consistency and interpretability consideration. We propose SHAP-PGD, an interpretable white-box attack method that combines global feature attribution with local gradient-based optimization. By leveraging global Shapley values for feature selection, performing refined optimization within a constraint-satisfying subspace, and utilizing a decoupled gradient masking mechanism, SHAP-PGD produces realistic perturbations under complex constraints and explains how these perturbations influence model decisions during the attack process. Finally, to quantitatively assess semantic consistency, we draw on synthetic distribution metrics and downstream model utility, providing a concrete and scalable formulation of the consistency problem. Experiments on four datasets demonstrate that SHAP-PGD advances adversarial attack for tabular data, providing new directions for model security in real-world settings.

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

# A APPENDIX

## A.1 ALGORITHM

The complete implementation process of the SHAP-PGD is shown in Algorithm 1:

---
**Algorithm 1** SHAP-PGD

---
**Require:** Classifier $h_\theta$, sample $x$, label $y$
**Require:** Budgets $(\epsilon, k)$, iterations $T$, step sizes $(\alpha, \beta)$
**Require:** SHAP refresh period $T_s$, stagnation patience $P$
**Ensure:** Adversarial example $x_{\mathrm{adv}}$
1: $\hat{g} \leftarrow \mathrm{SHAP}(x)$                 ▷ grad-SHAP via Eq. 8
2: $s^{(0)} \leftarrow \mathbf{0}$                 ▷ perturbation magnitudes
3: $stall \leftarrow 0$
4: $m^{(0)} \leftarrow \sigma(\hat{g})$             ▷ soft mask used only for gradients
5: **for** $t = 0$ **to** $T - 1$ **do**
6:      $x_{\mathrm{adv}}^{(t)} \leftarrow \mathrm{clip}_{[0,1]}\big(x + s^{(t)} \odot \Pi_0 m^{(t)}\big)$
7:      $\mathcal{L}^{(t)} \leftarrow \mathcal{L}\big(h_\theta, x_{\mathrm{adv}}^{(t)}\big)$        ▷ **update magnitude** $s$ (Eq. 11)
8:      $\widetilde{s}^{(t+1)} \leftarrow s^{(t)} + \alpha \, \mathrm{sign}\big(\nabla_{s^{(t)}} \mathcal{L}^{(t)}\big)$
9:      $s^{(t+1)} \leftarrow \Pi_{[-\epsilon,\epsilon]}\big(\widetilde{s}^{(t+1)}\big)$      ▷ **update location** $m$ (Eq. 12)
10:     $m^{(t+1)} \leftarrow m^{(t)} + \beta \dfrac{\nabla_{m^{(t)}} \mathcal{L}^{(t)}}{\|\nabla_{m^{(t)}} \mathcal{L}^{(t)}\|_2 + \varepsilon_{\mathrm{num}}}$     ▷ **stagnation detection**
11:     **if** $\mathrm{TopK}(m^{(t+1)}, k) == \mathrm{TopK}(m^{(t)}, k)$ **then**
12:        $stall \leftarrow stall + 1$
13:     **else**
14:        $stall \leftarrow 0$
15:     **end if**
16:     **if** $stall \geq P$ **and** attack fails **then**
17:        $m^{(t+1)} \leftarrow \mathrm{random\_init}(k)$
18:        $stall \leftarrow 0$
19:     **end if**                ▷ **periodic SHAP correction**
20:     **if** $(t + 1) \bmod T_s == 0$ **then**
21:        $\hat{g} \leftarrow \mathrm{SHAP}\big(x_{\mathrm{adv}}^{(t+1)}\big)$
22:        $m^{(t+1)} \leftarrow \hat{g}$
23:     **end if**
24: **end for**
25: **return** $x_{\mathrm{adv}}^{(T)}$

---

## A.2 COMPLEXITY ANALYSIS

Let $d$ denote the input dimensionality, $T$ the number of PGD iterations, $T_s$ the interval between SHAP refreshes, $|N|$ the number of background samples used by gradient SHAP, $m$ the number of Riemann steps along each IG path, and $C_{\mathrm{net}}$ the computational cost of a single network evaluation composed of both a forward and backward pass (i.e., the unit cost for obtaining $\nabla_x h_\theta$). Next, we analyze the complexity of Shapley and SHAP-PGD in turn.

**Shapley Complexity Analysis.** Computing exact Shapley values requires evaluating every possible feature coalition, leading to a complexity of $\mathcal{O}(2^d C_{\mathrm{net}})$, an exponential cost that quickly becomes impractical even for moderately sized $d$. Kernel-based or sampling-based SHAP reduces this to $\mathcal{O}(M C_{\mathrm{net}})$, where $M = \mathcal{O}(d \log d)$. Monte Carlo samples are typically insufficient for convergence. In contrast, our **white-box gradient SHAP** method (Eq. 8) computes an explanation by averaging IG over $|N|$ background samples, with $m$ steps taken along each straight-line path. Consequently, the computational cost of producing a single explanation is only $\mathcal{O}(|N| m d C_{\mathrm{net}})$.

**SHAP-PGD Complexity Analysis.** The total computational cost of SHAP-PGD comprises two components: (i) the PGD inner loop, and (ii) the periodic SHAP refreshes that provide global guidance. In the PGD inner loop, each iteration requires a single forward and backward pass through the network to obtain the gradient $\nabla_x h_\theta$. All subsequent vector operations—such as projection, element-wise updates, and stagnation checks—are linear in $d$ and thus negligible compared to the network computation. This means that each PGD step has a cost of $\mathcal{O}(C_{net})$, leading to $\mathcal{O}(T\, C_{net})$ over $T$ steps. For the SHAP refreshes, SHAP-PGD invokes the explainer once at initialization ($t = 0$), and then every $T_s$ iterations thereafter, for a total of $1 + \lceil T/T_s \rceil$ explanations. The computational cost per explanation is $\mathcal{O}(|N|\, m\, d\, C_{net})$, resulting in a total explanation cost of $\mathcal{O}\big((1 + \lceil T/T_s \rceil)\, |N|\, m\, d\, C_{net}\big)$. Summing both components, the overall complexity is

$$\mathcal{O}\Big( C_{net}\Big[ T \,+\, |N|\, m\, d\, (1 + \lceil T/T_s \rceil) \Big] \Big). \tag{14}$$

This shows that SHAP-PGD maintains a tractable complexity, introducing only a modest, tunable overhead compared to standard PGD, while benefiting from enhanced global explanatory guidance.

### A.3 THEORETICAL GUARANTEES

In this section, we provide a theoretical analysis showing that SHAP-PGD can generate realistic perturbations while strictly adhering to complex constraints. The proof proceeds in three steps:

**1. Reliability of the Global Shapley Guidance Phase.** We first establish why the SHAP-guided phase yields a consistent global importance estimate$c$ with exact $\phi_j(x)$. Assume each integrated path $\gamma_n(\eta) = z^{(n)} + \eta(x - z^{(n)})$ lies within the differentiable region of $h_y$. Define

$$\widehat{\phi}_j(x) = \frac{1}{|N|} \sum_{n=1}^{|N|} (x_j - z_j^{(n)}) \int_0^1 \frac{\partial h_y(\gamma_n(\eta))}{\partial x_j}\, d\eta. \tag{15}$$

Then, $\mathbb{E}_{z^{(n)} \sim \mathcal{P}_{data}}\big[\widehat{\phi}_j(x)\big] = \phi_j(x)$. This result follows from the exchangeability property of path-integrated gradients and the equivalence between Shapley values and integrated gradients (Sundararajan & Najmi, 2020). Hence, as $|N| \to \infty$, $\widehat{\phi}_j(x) \xrightarrow{p} \phi_j(x)$. For a finite number of samples, a Hoeffding bound gives

$$\Pr\big(|\widehat{\phi}_j - \phi_j| \geq \epsilon_{stat}\big) \leq 2 \exp\left( -\frac{2|N|\epsilon_{stat}^2}{B^2} \right), \tag{16}$$

where $B$ is an upper bound on the gradients and $\epsilon_{stat}$ is the statistical error tolerance. Thus, $\widehat{\phi}$ can reliably rank the most important features, mitigating the effect of gradient noise and reducing the risk of poor local optima during feature selection.

**2. Sufficient Descent in the Local PGD Subproblem.** Given the perturbation support $m$, the PGD subproblem efficiently determines the optimal perturbation amplitude $s$. With $m$ fixed, this reduces to solving $\min_{s \in [-\epsilon, \epsilon]^d} \mathcal{J}(x + s \odot m)$. Suppose $\mathcal{J}$ is $L$-smooth loss function (i.e., cross entropy loss) with respect to its input and $L$ is lipschitz–smooth constant. For step size $\alpha \leq \frac{\epsilon}{\sqrt{T}L}$, the projected gradient descent sequence is defined by

$$s^{(t+1)} = \Pi_{[-\epsilon, \epsilon]}\big(s^{(t)} - \alpha \nabla_{s^{(t)}} \mathcal{J}\big). \tag{17}$$

It then follows that

$$\min_{0 \leq t < T} \big\|\nabla_{s^{(t)}} \mathcal{J}\big\|_2^2 \leq \frac{2L\big(\mathcal{J}(x) - \mathcal{J}^\star(m)\big)}{T},$$

where $\mathcal{J}^\star(m)$ is the optimal value of the subproblem. Thus, as $T \to \infty$, the gradient norm converges to zero, $s^{(t)}$ approaches a first-order stationary point, and the constraint $||s|| \leq \epsilon$ always holds.

**3. Sequential Convergence of the Bilevel Alternating Optimization.** We next show that the alternating bilinear update framework over $(m, s)$ converges to a first-order stationary point and always respects the constraints. The algorithm alternates block coordinate descent over $(m, s)$:

- *Update $s$*: Perform PGD on the current support $m$.
- *Update $m$*: With $s$ fixed, perform a regularized gradient step, followed by a Top-K projection.

Table 7: Dataset Statistics

| Dataset | Domain | Output to flip | Total size | # Features | # Constraints | Balance (%) |
|---------|--------|----------------|-----------|-----------|--------------|-------------|
| CTU | Botnet detection | Malicious connections | 198 128 | 756 | 360 | 99.3/0.7 |
| LCLD | Credit scoring | Reject loan request | 1 220 092 | 28 | 9 | 80/20 |
| URL | Phishing | Malicious URL | 11 430 | 63 | 14 | 50/50 |
| WIDS | ICU survival | Expected survival | 91 713 | 186 | 31 | 91.4/8.6 |

The variable $m$ is restricted to be a $k$-sparse binary vector, yielding a finite state space of size $\binom{d}{k}$. Define $\mathcal{F}(m,s) = \mathcal{J}(x + s \odot m)$. We also introduce a penalty term $R(m) = \mathbb{I}[\|m\|_0 > k]$, leading to the overall objective:

$$\min_{m,s} \quad \mathcal{F}(m,s) + R(m) \quad \text{s.t. } s \in [-\epsilon, \epsilon]^d. \tag{18}$$

We follow the following assumptions: 1. Each PGD update ensures at least $\Delta_s > 0$ descent in $\mathcal{F}$ (or converges to a stationary point for the subproblem), where $\Delta_s$ is a minimum descent amount; 2. If $m$ does not improve over several iterations, a stagnation detection mechanism is triggered and $m$ is reset; 3. SHAP is periodically re-evaluated and Top-K features are chosen every $T_s$ steps. Then the sequence $\{(m^{(t)}, s^{(t)})\}$ produced by the algorithm must, after finitely many steps, enter a recurrent class in which $\mathcal{F}$ can no longer be improved (since the state space is finite). At this point:

(a) $\|m\|_0 \leq k$, $\|s\|_{\{2,\infty\}} \leq \epsilon$; thus, $\delta = s \odot m \in \mathcal{D}$, where $\mathcal{D}$ is the feasible region that satisfies the constraints $L_0$ and $L_{\{2,\infty\}}$.

(b) For all feasible $(m', s')$ differing from $(m, s)$ by only a single block ($m$ or $s$), further decrease in $\mathcal{F}$ is impossible—i.e., the block coordinate first-order optimality condition is satisfied.

Since the state space is finite and each successful descent strictly decreases the objective (which is bounded below), the number of descent steps is finite. Subsequently, the process must remain within a set of stationary points.

### A.4 EXPERIMENTAL PROTOCOL

**Datasets.** Consistent with previous work(Simonetto et al., 2024b;a), we evaluate our method on four public datasets; Table 7 summarizes their statistics.

- **URL.** This dataset is designed for phishing detection, where malicious URLs mimic legitimate sites to steal personal or financial information. Simonetto et al. (2024b;a) extract a variety of features—such as counts of specific substrings (e.g., "www", "&", ";"), URL length, port number, presence of brand names, as well as features from external services (e.g., Google indexing status, page rank, DNS presence)—for distinguishing phishing from benign URLs. Fourteen relational constraints are extracted, including seven linear constraints (e.g., hostname length $\leq$ total URL length) and seven Boolean constraints (e.g., if the count of "http" $> 0$ then the slash "/" count $> 0$).
- **LCLD.** Each record in this dataset represents an accepted loan, which is either fully repaid or charged off. Following Simonetto et al. (2024b;a), only samples with "loan status" in {Fully paid, Charged Off} are retained (binary classification). Features unavailable at origination or with $> 30\%$ missing values (in the training set), as well as redundant categorical features, are removed or aggregated; categorical variables are one-hot encoded. The final version contains 47 input features and 1 target variable, with feature boundary constraints set according to training set extrema. Nineteen Lending Club-controlled features are treated as immutable, and ten feature relations (three linear, seven nonlinear) are defined.
- **CTU.** The CTU-13 dataset (Chernikova & Oprea, 2022) contains labeled network traffic from a university, distinguishing between benign and botnet events. Raw network logs are aggregated by port. The data include 143,000 training and 55,000 test samples, with botnet traffic forming just 0.74%. Of 756 features, 432 are mutable. Researchers identify constraints both on connection counts (to prevent their artificial reduction) and on protocol properties (e.g., TCP/UDP single-packet maximum size of 1500 bytes), resulting in 360 constraints.
- **WIDS.** Proposed by Lee et al. (2020), this dataset contains critical care survival data for ICU patients. The prediction target is patient survival, given biological and clinical features; the data are highly imbalanced. Thirty linear relational constraints are included.

**Architectures.** We focus on the most widely used deep learning models for tabular data:

- **TabTransformer** (Huang et al., 2020) leverages self-attention to transform categorical features into contextual embeddings, enhancing both interpretability and robustness to noisy inputs.
- **TabNet** (Arik & Pfister, 2021) employs a multi-step decision process with sequential attention, dynamically selecting features at each stage and aggregating information for prediction.
- **RLN** (Regularization Learning Networks) (Shavitt & Segal, 2018) minimizes counterfactual loss by learning regularization coefficients for network weights, resulting in highly sparse models and reduced sensitivity.
- **STG** (Stochastic Gates) (Yamada et al., 2020) performs neural network feature selection via a stochastic gating mechanism, implementing a probabilistic relaxation of the feature $l_0$-norm.
  **VIME** (Value Imputation for Mask Estimation) (Yoon et al., 2020) combines deep encoders and predictors with self-supervised and semi-supervised learning for feature imputation and completion.

**Baselines.** We compare SHAP-PGD with state-of-the-art adversarial attack methods that satisfy $L_0$ and $L_{\{2,\infty\}}$ constraints:

- **PGD$_0$** (Croce & Hein, 2019) is a gradient-based method for generating adversarial examples under $L_0$ norm constraints. It greedily selects and modifies the most influential features to minimize the number of changes, enabling effective attacks with sparse feature modifications.
- **Sparse-PGD** (Zhong et al., 2024) employs a sparse projected gradient descent strategy, also targeting $L_0$ sparsity constraints. By optimizing the projection step during gradient updates, it enhances both the attack strength and sparsity of adversarial examples, making it applicable to settings with limited modifications to continuous features.
- **CaFA** (Ben-Tov et al., 2024) introduces a constrained and fast adversarial generation algorithm that efficiently optimizes perturbations under $L_0$ and $L_{2,\infty}$ constraints. It balances attack effectiveness and perturbation controllability, and is suitable for tabular data and high-dimensional continuous feature spaces.
- **CAPGD** (Simonetto et al., 2024a) is a projection-based gradient descent framework that projects adversarial perturbations onto specific norm sets (such as $L_0$ or $L_{2,\infty}$), systematically enhancing attack interpretability and effectiveness. It is adaptable to adversarial generation tasks under various perturbation constraints.
- **BF*** (Kulynych et al., 2018) is a heuristic search-based attack method for continuous features, exhaustively attempting all Boolean changes for each feature to identify effective attack paths with minimal perturbation, and is often used for evaluating model robustness to continuous inputs.
- **MOEVA** (Simonetto et al., 2021) leverages a multi-objective evolutionary algorithm framework to optimize adversarial perturbations. It automatically searches for optimal adversarial examples under multiple constraints in the continuous feature space, balancing perturbation cost and attack success rate.

**Hyperparameter Settings.** For BF*, CAPGD, and MOEVA, we adopt the hyperparameters recommended in their original papers. For PGD$_0$, we set the step size to $\eta = 1.25$, and each attack runs for 200 iterations. Additionally, we use 5 restarts to further enhance performance. For Sparse-PGD, both step sizes $\alpha$ and $\beta$ are set to 0.1, with 200 iterations and a tolerance of 3. For SHAP-PGD, the SHAP cycle update value $T_s$ is set to 10, both $\alpha$ and $\beta$ are 0.1, the number of iterations is 40, and the tolerance is 3. In the adversarial training, we keep the default parameters of the architecture model and set the iteration of the adversarial attack to 40.

**Metrics.** We introduce the indicators involved in the article in detail:

- **Robust Accuracy.** The accuracy on valid adversarial examples generated by a given attack. Misclassified clean examples are not altered. If an adversarial example is invalid, it is counted as correctly classified.
- **Jensen-Shannon Divergence (JSD).** JSD quantifies the similarity between two probability distributions. As a symmetrized and normalized version of KL divergence, its value ranges in $[0, 1]$, where a smaller value indicates higher similarity. For distributions $P$ and $Q$,

$$\text{JSD}(P \parallel Q) = \frac{1}{2}D_{\text{KL}}(P \parallel M) + \frac{1}{2}D_{\text{KL}}(Q \parallel M), \tag{19}$$

where $M = \frac{1}{2}(P + Q)$. $D_{\text{KL}}$ denotes Kullback-Leibler divergence: $D_{\text{KL}}(P \parallel Q) = \sum_x P(x) \log\left(\frac{P(x)}{Q(x)}\right)$.

- **Wasserstein Distance ($d_{\text{Wass.}}$).** Also known as Earth Mover's Distance, it measures the minimal "cost" to transport mass between two probability distributions, with a smaller value indicating

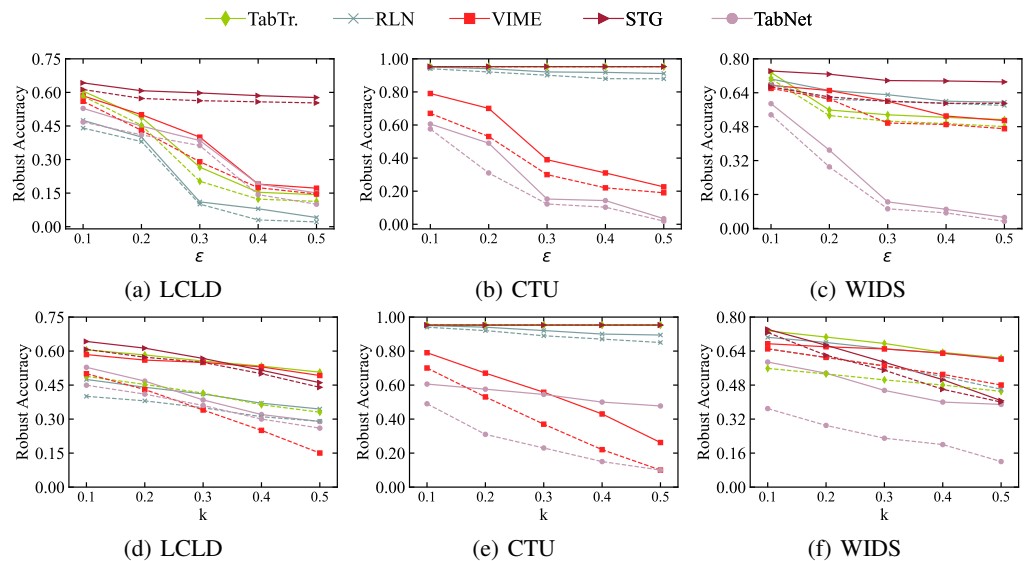

Figure 5: The impact of perturbation budget on robust accuracy for other three datasets

Table 8: Transfer attack between STG, TabNet and TabTr.

| Method | Dataset | TabTr. → STG | TabTr. → TabNet | TabNet → STG | TabNet → TabTr. | STG → TabTr. | STG → TabNet |
|---|---|---|---|---|---|---|---|
| SHAP-PGD | URL | 9.5 | 3.9 | 20.1 | 9.8 | 21.7 | 20.1 |
| | LCLD | 53.9 | 8.5 | 55.9 | 38.8 | 30.4 | 10.0 |
| | CTU | 95.3 | 0.4 | 95.3 | 95.3 | 95.3 | 5.5 |
| | WIDS | 58.8 | 0.2 | 61.2 | 51.3 | 67.8 | 28.65 |
| Sparse-PGD | URL | 14.5 | 8.9 | 23.1 | 12.8 | 16.7 | 25.2 |
| | LCLD | 68.9 | 23.5 | 70.9 | 53.8 | 45.4 | 28.4 |
| | CTU | 95.3 | 18.3 | 95.3 | 95.3 | 95.3 | 10.3 |
| | WIDS | 73.8 | 15.2 | 73.2 | 61.3 | 70.8 | 43.6 |

greater similarity. Given one-dimensional distributions $P$ and $Q$,

$$d_{\text{Wass.}}(P, Q) = \inf_{\gamma \in \Gamma(P,Q)} \int_{X \times Y} |x - y| \, d\gamma(x, y), \tag{20}$$

where $\Gamma(P, Q)$ is the set of all joint distributions with marginals $P$ and $Q$.

- **Beta Score ($\beta$-Score).** The $\beta$-Score evaluates how well synthetic data reproduces the joint feature distribution of real data, based on the classifier two-sample test. It is defined as:

$$\beta\text{-Score} = 1 - 2 \cdot |\text{AUC} - 0.5|, \tag{21}$$

where AUC is the area under the ROC curve of a discriminator distinguishing real from synthetic samples. A value closer to 1 indicates that the two datasets are harder to separate.

- **Coverage Score ($C$-Score).** The $C$-Score assesses whether the synthetic data "covers" the space of real data, preventing missing modes. It calculates the fraction of real data points for which their nearest synthetic sample is within a distance threshold $\epsilon$:

$$C\text{-Score} = \frac{1}{|X|} \sum_{x \in X} \mathbb{I}\left(\min_{y \in Y} d(x, y) < \epsilon\right), \tag{22}$$

where $X$ and $Y$ are the sets of real and synthetic samples, respectively, $d(\cdot, \cdot)$ is a distance metric, and $\mathbb{I}(\cdot)$ is the indicator function.

**Hardware.** All experiments were conducted on a High-Performance Computing cluster. Each compute node is equipped with two AMD EPYC 7H12 CPUs (a total of 128 cores @ 2.6 GHz) and 256 GB of RAM. For each experimental run, we allocated 32 CPU cores and 64 GB of RAM.

A.5 IMPACT OF ATTACK BUDGET

In this section, we further analyze the impact of $\epsilon$ and $k$ on the LCLD, CTU, and WIDS datasets. As shown in Fig.5, the solid and dashed lines represent $k = 0.1$ and $k = 0.2$ under varying $\epsilon$. When $\epsilon$

Table 9: Robust accuracy with subset of constraints under $L_2 = 0.5$. $\Omega$ is the complete set of constraints. For CG3, we additionally evaluate with 10%, 25%, and 50% of the group.

| Group | | CAPGD | | | | | SHAP-PGD | | | | |
|---|---|---|---|---|---|---|---|---|---|---|---|
| | | RLN | STG | TabNet | TabTr. | VIME | RLN | STG | TabNet | TabTr. | VIME |
| $\Omega$ | | 97.8 | 95.3 | 96.1 | 95.3 | 95.1 | 88.9 | 95.3 | 0.0 | 95.3 | 38.4 |
| Ablation | $\Omega \setminus$ CG0 | 97.8 | 95.3 | 96.1 | 95.3 | 95.1 | 88.9 | 95.3 | 0.0 | 95.3 | 38.4 |
| | $\Omega \setminus$ CG1 | 97.8 | 95.3 | 96.1 | 95.3 | 95.1 | 88.9 | 95.3 | 0.0 | 95.3 | 38.4 |
| | $\Omega \setminus$ CG2 | 97.8 | 95.3 | 96.1 | 95.3 | 95.1 | 87.7 | 95.3 | 0.0 | 95.3 | 37.6 |
| | $\Omega \setminus$ CG3 | 97.8 | 95.3 | 96.1 | 95.3 | 95.1 | 86.6 | 95.3 | 0.0 | 95.3 | 38.3 |
| Components | CG0 | 97.8 | 95.3 | 96.1 | 95.3 | 95.1 | 84.7 | 95.3 | 0.0 | 95.3 | 37.4 |
| | CG1 | 97.8 | 95.3 | 96.1 | 95.3 | 95.1 | 84.3 | 95.3 | 0.0 | 95.3 | 36.5 |
| | CG2 | 75.3 | 95.3 | 31.4 | 94.3 | 0.0 | 86.1 | 95.3 | 0.0 | 95.3 | 38.1 |
| | CG3 | 97.2 | 95.3 | 95.3 | 95.3 | 95.1 | 87.2 | 95.3 | 0.0 | 95.3 | 37.6 |
| Percentage CG3 | 10% | 85.2 | 95.3 | 43.0 | 95.1 | 11.3 | 77.6 | 95.3 | 0.0 | 95.3 | 10.0 |
| | 25% | 93.4 | 95.3 | 59.9 | 95.3 | 37.8 | 83.3 | 95.3 | 0.0 | 95.3 | 27.4 |
| | 50% | 94.9 | 95.3 | 84.5 | 95.3 | 93.2 | 86.9 | 95.3 | 0.0 | 95.3 | 35.8 |

Table 10: Robust accuracy with subset of constraints under $L_0 = 0.1 \& L_2 = 0.5$. $\Omega$ is the complete set of constraints. For CG3, we additionally evaluate with 10%, 25%, and 50% of the group.

| Group | | Sparse-PGD | | | | | SHAP-PGD | | | | |
|---|---|---|---|---|---|---|---|---|---|---|---|
| | | RLN | STG | TabNet | TabTr. | VIME | RLN | STG | TabNet | TabTr. | VIME |
| $\Omega$ | | 96.2 | 95.3 | 4.6 | 95.3 | 44.5 | 91.1 | 95.3 | 3.4 | 95.3 | 22.6 |
| Ablation | $\Omega \setminus$ CG0 | 96.2 | 95.3 | 4.6 | 95.3 | 44.5 | 91.1 | 95.3 | 3.4 | 95.3 | 22.6 |
| | $\Omega \setminus$ CG1 | 96.2 | 95.3 | 4.6 | 95.3 | 44.5 | 91.1 | 95.3 | 3.4 | 95.3 | 22.6 |
| | $\Omega \setminus$ CG2 | 95.7 | 95.3 | 4.1 | 95.3 | 40.4 | 90.6 | 95.3 | 2.9 | 95.3 | 21.9 |
| | $\Omega \setminus$ CG3 | 96.0 | 95.3 | 3.8 | 95.3 | 42.7 | 89.9 | 95.3 | 3.1 | 95.3 | 21.2 |
| Components | CG0 | 95.1 | 95.3 | 3.6 | 95.3 | 39.2 | 86.2 | 95.3 | 1.1 | 95.3 | 18.7 |
| | CG1 | 95.5 | 95.3 | 3.5 | 95.3 | 40.1 | 85.5 | 95.3 | 2.0 | 95.3 | 19.8 |
| | CG2 | 94.1 | 95.3 | 3.5 | 95.3 | 40.8 | 89.4 | 95.3 | 2.5 | 95.3 | 20.3 |
| | CG3 | 94.8 | 95.3 | 2.9 | 95.3 | 39.6 | 88.3 | 95.3 | 2.8 | 95.3 | 21.4 |
| Percentage CG3 | 10% | 87.4 | 95.3 | 1.1 | 95.3 | 28.4 | 79.7 | 95.3 | 0.5 | 93.3 | 13.4 |
| | 25% | 90.7 | 95.3 | 1.9 | 95.3 | 31.3 | 83.4 | 95.3 | 1.4 | 94.8 | 18.6 |
| | 50% | 94.2 | 95.3 | 2.5 | 95.3 | 33.8 | 86.9 | 95.3 | 2.7 | 95.2 | 20.9 |

increases from 0.1 to 0.5, the average accuracy drops by 44.0% for $k = 0.1$ and by 46.1% for $k = 0.2$, indicating that increasing the proportion of perturbed features only adds about 2% of additional accuracy loss. Thus, model vulnerability is mainly driven by perturbation magnitude $\epsilon$. Consistent with earlier results, a sharp decline in robustness occurs as $\epsilon$ increases from 0.3 to 0.4, with an average drop of 49.8%. Similarly, for different $k$ values (solid for $\epsilon = 0.1$, dashed for $\epsilon = 0.2$), increasing $k$ leads to an approximately linear decrease in robustness, with each additional 10% of perturbed features reducing robust accuracy by an average of 8.4%.

## A.6 COMPLEX CONSTRAINTS ANALYSIS

Previous experiments revealed that all attack algorithms achieve low success rates on the CTU dataset, mainly due to the presence of numerous complex feature constraints, which make it difficult for adversarial examples to simultaneously satisfy all requirements. To further investigate the impact of feature constraints on SHAP-PGD, we conducted in-depth experiments on the CTU dataset. Following prior work (Simonetto et al., 2024a), we categorize the CTU constraints by complexity:

• **CG0:** A single constraint involving 90 features, $\sum_i F_i = \sum_j F_j$, where both terms represent the total number of sent packets.

Table 11: Evaluation at runtime. Lower runtime (in seconds) indicates a more effective attack. We highlight the lowest runtime in **bold**. Following prior work, we set $L_2 = 0.5$ and $L_0 = 0.1$.

| Dataset | Model | $L_2 = 0.5$ | | | | $L_2 = 0.5 \& L_0 = 0.1$ | | | |
|---------|-------|-------|-----|-------|----------|---------|-----------|------|----------|
| | | CAPGD | BF* | MOEVA | SHAP-PGD | $PGD_0$ | Sparse-PGD | CaFA | SHAP-PGD |
| URL | TabTr. | **1** | 33 | 75 | 83 | 8 | **4** | 35 | 91 |
| | RLN | **1** | 27 | 74 | 80 | 10 | **6** | 40 | 85 |
| | VIME | **2** | 32 | 70 | 76 | 10 | **1** | 33 | 72 |
| | STG | **2** | 58 | 90 | 98 | 12 | **6** | 45 | 85 |
| | TabNet | **8** | 444 | 165 | 174 | **3** | 15 | 75 | 177 |
| LCLD | TabTr. | **5** | 154 | 124 | 135 | 7 | **1** | 55 | 130 |
| | RLN | **1** | 147 | 50 | 58 | 7 | **3** | 25 | 55 |
| | VIME | **1** | 149 | 49 | 53 | 10 | **4** | 22 | 49 |
| | STG | **3** | 191 | 60 | 67 | **1** | 11 | 28 | 78 |
| | TabNet | **4** | 754 | 68 | 74 | 10 | **1** | 31 | 83 |
| CTU | TabTr. | **4** | 371 | 98 | 109 | 9 | **2** | 49 | 125 |
| | RLN | 9 | **12** | 98 | 106 | **1** | 18 | 51 | 113 |
| | VIME | **4** | 924 | 107 | 115 | 11 | **5** | 50 | 106 |
| | STG | **5** | 548 | 105 | 112 | **1** | 15 | 53 | 139 |
| | TabNet | **7** | 816 | 157 | 169 | 10 | **9** | 79 | 143 |
| WIDS | TabTr. | **3** | 440 | 65 | 72 | 7 | **1** | 33 | 71 |
| | RLN | **3** | 2520 | 52 | 59 | 12 | **8** | 26 | 65 |
| | VIME | **2** | 1406 | 48 | 55 | 10 | **3** | 24 | 58 |
| | STG | **3** | 1888 | 64 | 71 | **1** | 9 | 32 | 62 |
| | TabNet | **5** | 10472 | 77 | 85 | 10 | **8** | 38 | 81 |

- **CG1:** A single constraint involving 90 features, $\sum_i F_i = \sum_j F_j$, where both terms represent the total number of received packets.
- **CG2:** 34 constraints of the form BYTE/PACKETS $\leq 1500$, reflecting the maximum bytes per packet.
- **CG3:** 324 constraints of the form $A \leq B$, where $A$ and $B$ denote port- and direction-specific statistics.

In our experiments, we first evaluate robust accuracy when each group of constraints is omitted in turn. Next, we assess SHAP-PGD performance when applying each group individually. Finally, we report how the proportion of CG3 constraints affects the robust accuracy. As shown in Table 9, we first evaluate CAPGD and SHAP-PGD under an $L_2 = 0.5$ constraint. The results indicate that CAPGD is largely insensitive to the removal of constraints and is almost ineffective in performing successful attacks. In contrast, after removing constraints CG2 and CG3, SHAP-PGD achieves average improvements in attack robust accuracy of 2.3% and 0.8% on RLN and VIME, respectively. Moreover, adjusting the coverage ratio of CG3 significantly impacts algorithm performance: when only 10% of the CG3 constraints are retained, the robust accuracy of VIME drops sharply from 37.6% to 10.0%; as the proportion of CG3 constraints increases, robust accuracy gradually recovers, with a maximum improvement of up to 27.6% percentage points (from 10.0% to 37.6%). Similarly, we further analyze model performance under the joint constraint of $L_0 = 0.1$ and $L_2 = 0.5$. Experimental results show that, when different constraints are removed, Sparse-PGD achieves an average improvement in attack robust accuracy of 1.2%, which is lower than the 2.4% achieved by SHAP-PGD. Moreover, comparing different constraint configurations reveals that the performance gain under the joint $L_0 = 0.1 \& L_2 = 0.5$ constraint is greater than that under the single $L_2 = 0.5$ constraint (the improvement increases from 1.5% to 2.4%). These results suggest that joint constraints more accurately reflect the complexity and challenges of real-world applications, and thus are of greater practical significance.

## A.7 TIME EFFICIENCY ANALYSIS

Table 11 shows that CAPGD and Sparse-PGD are, on average, roughly $\times 15$ faster than SHAP-PGD. The bottleneck for SHAP-PGD is the periodic computation of Shapley values, whose per-sample complexity is $\mathcal{O}(|N| \, m \, d \, C_{\text{net}})$ (see Appendix A.2). To accelerate this step, one could replace the neural network explainer with a tree-based surrogate $g(\cdot)$ (e.g., GBDT, Random Forest, XGBoost) and estimate feature attributions via TreeSHAP (Lundberg et al., 2018). For a single instance, TreeSHAP requires only $\mathcal{O}(B \, L \, D)$ time, where $B$ is the number of trees, $L$ the average leaf count, and $D$ the maximum depth—orders of magnitude cheaper than $\mathcal{O}(|N| \, m \, d \, C_{\text{net}})$ for large $d$ or deep networks. However, we do not adopt this surrogate for two reasons: **(i)** Preliminary experiments show that

Table 12: Evolution for robust accuracy on the URL and LCLD datasets (The same trend is seen in the other two datasets) for each component and variant of SHAP-PGD.

| Ablations | URL | | | | | LCLD | | | | |
|---|---|---|---|---|---|---|---|---|---|---|
| | TabTr. | RLN | VIME | STG | TabNet | TabTr. | RLN | VIME | STG | TabNet |
| SHAP-PGD | 3.7 | 10.4 | 22.2 | 17.6 | 8.9 | 14.3 | 4.1 | 17.2 | 57.7 | 15.1 |
| SHAP-PGD w/o GSG | 6.2 | 18.4 | 33.3 | 19.6 | 10.4 | 14.3 | 8.7 | 28.1 | 59.9 | 17.6 |
| SHAP-PGD w/o DGM | 24.7 | 20.0 | 48.4 | 26.3 | 19.5 | 18.1 | 5.8 | 27.9 | 58.1 | 42.3 |
| SHAP-PGD w/o SD | 3.9 | 12.7 | 22.7 | 17.6 | 10.3 | 14.3 | 8.2 | 17.2 | 60.8 | 17.4 |
| SHAP-PGD w/o PC | 4.5 | 17.8 | 25.1 | 21.5 | 14.2 | 20.4 | 11.7 | 17.8 | 62.8 | 18.8 |
| SHAP-PGD w $GSG_{all}$ | 3.7 | 9.6 | 15.8 | 17.6 | 7.7 | 10.7 | 4.1 | 17.2 | 57.3 | 14.0 |
| SHAP-PGD w ASS | 4.4 | 10.4 | 22.2 | 18.7 | 9.1 | 16.5 | 7.5 | 17.9 | 58.6 | 20.3 |
| SHAP-PGD w MU | 3.3 | 13.2 | 22.6 | 25.7 | 8.2 | 20.8 | 3.3 | 20.4 | 62.7 | 15.2 |
| SHAP-PGD w DLR | 12.5 | 20.3 | 27.8 | 19.9 | 10.4 | 17.0 | 6.4 | 16.9 | 60.8 | 16.9 |

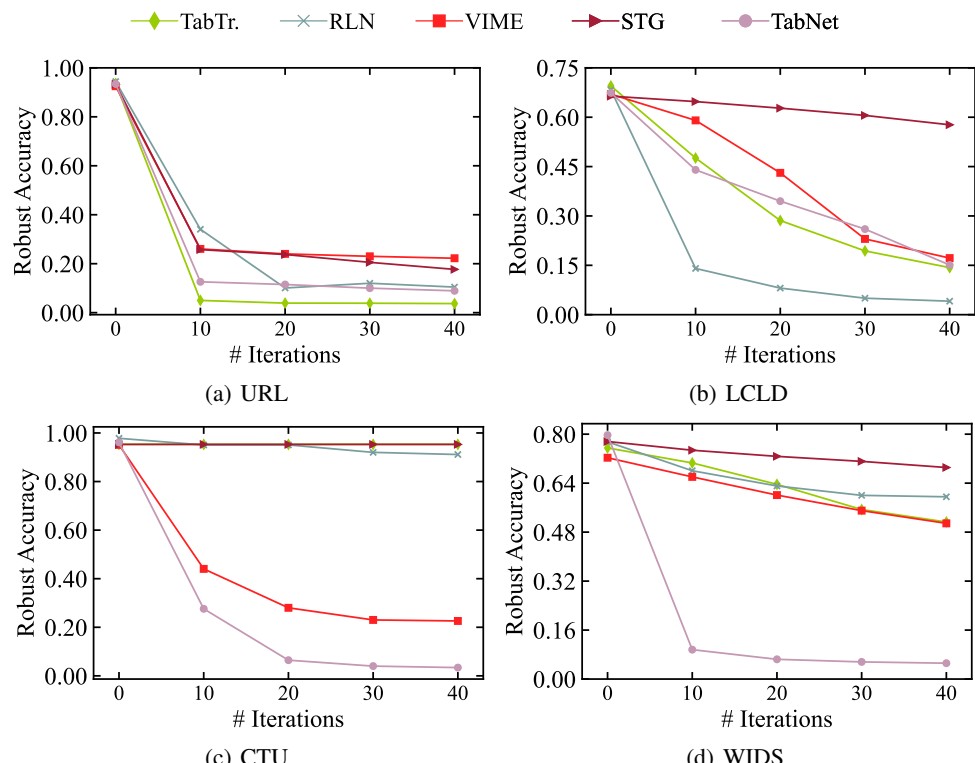

Figure 6: Evaluating robust accuracy for different numbers of iterations of SHAP-PGD

attacks guided by TreeSHAP yield noticeably higher robust accuracy than SHAP-PGD, indicating a non-trivial trade-off between speed and efficacy. **(ii)** Training an additional tree ensemble incurs its own computational overhead, which offsets part of the savings and complicates deployment pipelines. Given these drawbacks, we retain the original SHAP-PGD formulation and instead report its efficiency separately, leaving surrogate-based variants to future work aimed at applications with even tighter latency constraints.

## A.8 ABLATION STUDY

In this section, we systematically investigate the impact of global sharpley guidance and local gradient optimization on the performance of SHAP-PGD. As show in Table 12, We additionally analyze several variants, including popular PGD-based optimization techniques used under $L_2$ constraints, such as adaptive step size (ASS), momentum updates(MU), and difference of logits ratio (DLR).

**SHAP-PGD w/o GSG:** This variant evaluates the effect of removing the Global Sharpley Guidance Layer (GSG) from SHAP-PGD. Specifically, feature selection is based on the absolute value of the gradient instead of Shapley values. Experimental results on two datasets show that omitting GSG leads to a performance drop—an average decrease of 4.5% and up to 11.1%. This is because relying solely on gradients for feature selection tends to repeatedly update in the gradient direction, making the algorithm more prone to local optima.

**SHAP-PGD w/o DGM:** This variant assesses the impact of eliminating the Decoupled Gradient Mechanism (DGM). In this setting, the position vector $m$ is no longer updated, and $\delta^{t+1}$ is directly taken as the perturbation. Results show a pronounced performance degradation, with an average decrease of 12.0% and a maximum of 27.2%. The absence of $m$ updates leads to repeated perturbation of the same set of features throughout iterations, causing the attack to be trapped in local optima and missing features that should have been perturbed.

**SHAP-PGD w/o SD:** This variant evaluates the effect of removing Stagnation Detection (SD). The experiment shows an average drop in performance of 1.4% and a maximum of 4.1%, which is less severe than the previous two variants. This is because, under ideal conditions, perturbations are adequately updated in each iteration, so the overall impact on model performance is limited.

**SHAP-PGD w/o PC:** This analysis considers the effect of removing the Periodic Correction (PC) mechanism. Experiments demonstrate an average performance decrease of 4.3% and a maximum of 7.6%. Distinct from SD, PC heuristically recalibrates the position vector $m$ using Shapley values every few steps to help escape local optima. Thus, removing PC leads to a more noticeable decline in performance.

**SHAP-PGD w/o GSG$_{all}$.** Unlike the previous variant, which removes the global sharpley guidance entirely, here we modify the periodic global sharpley guidance to apply at every step (i.e., set $T_s = 1$). Experimental results show that more frequent guidance indeed enhances the attack performance of SHAP-PGD, with an average improvement of 8.3% and up to 28.8% in the best case. However, this increased guidance frequency incurs significant computational overhead. Considering both efficiency and effectiveness, we ultimately retain the periodic guidance setting for $T_s$.

**SHAP-PGD w AAS/MU/DLR:** Following prior work (Croce & Hein, 2020; Simonetto et al., 2024a), we incorporate adaptive step size (AAS), momentum update (MU), and difference of logits ratio (DLR) into the SHAP-PGD framework to evaluate their effectiveness under various constraints. Previous works demonstrate that these optimization strategies perform well under $L_{\{2,\infty\}}$ constraints, but their benefits are limited when jointly applying $L_0$ and $L_{\{2,\infty\}}$ constraints. Although there are marginal improvements in certain metrics, the overall trend is a decrease in performance, with an average drop of 10.0% and a maximum of 19.5%. Effectively integrating these methods for scenarios with complex constraints remains an open direction for future research.

## A.9 Hyperparameters Study

We compare the attack performance of SHAP-PGD under different numbers of iterations. As show in Fig. 6, it is evident that as the number of iterations increases, the robust accuracy steadily improves and eventually converges. Notably, SHAP-PGD demonstrates high attack efficiency. For instance, on the URL dataset, model performance drops rapidly from 96% to approximately 10% within just 10 iterations; similar trends are observed on the LCLD and WIDS datasets. The only exception is the CTU dataset, where the decrease is more gradual, mainly due to the presence of numerous complex feature constraints. Overall, these results indicate that SHAP-PGD can efficiently and effectively compromise models within a short computation time.

## A.10 The Use Of Large Language Models

In developing this article, we utilized artificial intelligence assistants to improve academic writing and debug code. These tools function similarly to human editors or code checkers, primarily improving clarity, grammatical correctness, and efficiency. All conceptual insights, technical arguments, and critical analysis are the original contributions of the authors.

