# OpenReview forum: "SHAP-PGD: A Realistic Adversarial Attack on Tabular Data by Unifying Interpretability and Semantic Consistency"
_ICLR.cc/2026/Conference — Submitted to ICLR 2026_

### Official Review · Reviewer_ED7q · 2025-10-20

**Soundness:** 2
**Presentation:** 1
**Contribution:** 1
**Rating:** 2
**Confidence:** 5

**Summary:**

This paper proposes SHAP-PGD, an interpretable white-box tabular adversarial attack under complex constraints. SHAP-PGD utilizes global attribution to identify the most influential features and uses a decoupled gradient masking mechanism within this selected constraint-satisfying subspace to avoid local minima.

**Strengths:**

1. The method is easy to understand
2. Extensive experiments suggest the effectiveness of the proposed method.

**Weaknesses:**

1. **Poor contribution:** The only contribution of this work is proposing an initialization method with prior knowledge about tabular data, which also introduces large computational overhead (indicate in Table 11). Apart from this, most algorithm designs are completely the same as Sparse-PGD [1], and the authors do not cite this paper in the method section.
2. **Too strong claim:** The authors claim the attack is interpretable and semantic consistent, but this is just a post-hoc observation. There is no technique in the algorithm to guarantee these properties. Although SHAP can find critical features before the attack, but how can you guarantee that the updated mask is still interpretable? To demonstrate the effectiveness of SHAP, you need to plot Sparse-PGD correlation in Figure 1.
3. **Unfair experimental settings:** Table 11 indicates that SHAP-PGD is 15x slower than baselines, e.g., CAPGD and Sparse-PGD. If you increase the computational budget of these methods to the the same level as your method, how will the results look like?


[1] Zhong, Xuyang, Yixiao Huang, and Chen Liu. "Towards Efficient Training and Evaluation of Robust Models against $ l_0 $ Bounded Adversarial Perturbations." Forty-first International Conference on Machine Learning. 2024.

**Questions:**

1. Your attack decouples perturbation magnitudes and locations to generate sparse perturbations. How do you generate $l_2$-bounded perturbations? Does your attack degrade to naive PGD in this case?
2. Why the result of Sparse-PGD is not included in Table 5?
3. Do you try to adversarially train models by using Sparse-PGD and SHAP-PGD? What's the implementation details of adversarial training?
4. Typos : What is "Sarse-PGD" at Table 2? At the 6th row of Table 4, why 30.1 is bold while 29.8 is smaller than 30.1?

---

### Official Review · Reviewer_XArW · 2025-10-27

**Soundness:** 3
**Presentation:** 2
**Contribution:** 3
**Rating:** 4
**Confidence:** 3

**Summary:**

This paper introduces a white-box adversarial attack method, SHAP-PGD, aiming for tabular data.

To respect real-world rules and logical relations in tabular data, the proposed method SHAP-PGD leverages the idea of Shapley values to identify the most important features for misclassification, and then conducts PGD only within the selected features to guide the perturbation.

Furthermore, the paper employs four synthetic data metrics (i.e., JSD, Wasserstein Distance, $\beta$-score, and $C$-score) to measure the distance between the distributions of original and adversarial samples, and show the proposed method is the one who generates adversarial examples that closely align with the original distribution (hence preserve the original semantics).

Experiments are conducted on four tabular datasets and show the effectiveness of the proposed method.

**Strengths:**

- The motivation is clear, and the proposed method (based on Shapley values) seems novel in the tabular domain.
- The work uses metrics to measure the semantic consistency, which seems important.
- Experimental results seem to be effective.

**Weaknesses:**

- Notations are confusing and need further clarity.
- The proposed method is about 15$\times$ slower than baselines, as well as the white-box assumption, which might limit the practicality. What does the method perform if considering a transfer attack?

**Questions:**

- In line 102, feature space is denoted as $F=$ \{ $f_1, f_2, ..., f_d$ \} , then input $x$ is obtained from $z$, so does this mean that $x$ is the feature and $x \in F$? What are the dimensions of $x$ and $f_i$? In Eq.(2), $f_i$ and $x$ denote the current feature value and original value, respectively; does this mean that $f_i = x + \delta$? Eq.(2) seems to be the constraints in the feature space, and needs some explanation about this equation. In addition, in Eq.(8), feature $x - z$ where $z$ is the training sample (object), how can it be operated for feature and object?
- What is the penalty function used in Eq.(5)?
- Eq.(12) updates the mask, then stagnation detection and periodic correction are adopted to avoid being trapped in a local optimum. Could the authors explain why not simply compute SHAP attributions every step, since in this case, the mask could be updated as well?
- For adversarial training, could the authors clarify why the mean clean accuracy increases?
- In line 17 of Algorithm 1, would different initializations affect the results?
- What is the impact of $T_s$?
- The proposed method identifies the features based on Shapley values. What if the estimation has errors? Would that significantly affect the results?

---

### Official Review · Reviewer_LosV · 2025-11-01

**Soundness:** 1
**Presentation:** 1
**Contribution:** 2
**Rating:** 2
**Confidence:** 3

**Summary:**

The paper proposes a new method for creating tabular data attacks. They propose a solution that allows them to generate attacks that minimally change wrt the original while also satisfying a set of complex constraints.

**Strengths:**

The paper introduces a nice idea which might be useful in practice. The paper though needs more refinement before being ready for publication.

**Weaknesses:**

The paper often does not introduce all the necessary notation or is written in a very cursory way. This makes it often very imprecise and hard to follow. Below I give some examples:

1. The loss function $ \mathcal{L} $ appears in equation (10) but is never defined in the text. Earlier, equation (5) defines $ \mathcal{L}(\theta, x+\delta) $. however, it is unclear whether these are identical to not.
2. Additionally, in equation (5) we have the term  $ \mathrm{penalty}(x,\omega) $, but no explicit form is provided (e.g., hinge, squared, barrier). The paper states that it “follows CPGD,” but does not restate the mapping from a constraint $ \omega $ to a differentiable penalty. Since constraint satisfaction is a central claim, the exact functional form should be included.
3. Equation (10) applies $ \Pi_0 $ to enforce the $ L_0 $ constraint, yet the paper does not specify whether this is hard top-$k$ thresholding, continuous relaxation, or another operator. Because the mask $ m^{(t)} $ becomes continuous in equation (12), the projection rule must be stated explicitly.
4. The mask is first defined via SHAP ranking (equation (9)), but later updated by gradient ascent and projected by $ \Pi_0 $. It is unclear whether $ K $ in equation  (9) equals the $ L_0 $ budget, whether sparsity is fixed, or whether $ \Pi_0 $ can override the SHAP selection. This affects both sparsity guarantees and comparability with baselines.


Regarding the experimental analysis, the authors use a single perturbation budget $(L_2 = 0.5,\ L_0 = 0.1)$ without sensitivity analysis.
These values are inherited from prior work but are arbitrary. It could be nice to have a plot there different perturbation budgets are considered.


More in general, the paper claims that constraints are “always satisfied in practice,” but no violation statistics, penalty magnitudes, or feasibility ratios are reported in tables.  In the main section of the paper they have a section "Theoretical Guarantees” which simply reads: "We provide theory to show that SHAP-PGD can generate realistic adversarial perturbations while satisfying complex constraints. See the AppendixA.3 for details". The appendix also contains proof sketches rather than guarantees.

**Questions:**

See weaknesses.

---

### Official Review · Reviewer_dvqL · 2025-11-03

**Soundness:** 3
**Presentation:** 3
**Contribution:** 2
**Rating:** 6
**Confidence:** 5

**Summary:**

This paper tackled the problem of adversarial attacks on tabular data, which differs from the standard problem in that attacks have to consider validity constraints between the features. To this aim, the paper proposes SHAP-PGD, an interpretable white-box adversarial attack for tabular data that unifies SHAP-based feature attribution with projected gradient descent (PGD) under complex constraints (immutability, boundaries, types, and feature relationships). The novelty of SHAP-PGD compared to other tabular adversarial attacks lies in the use of SHAP to pre-select promising features to modify. Through experiments on four real-world tabular datasets (URL, LCLD, CTU, WIDS) and five models (TabTransformer, TabNet, RLN, STG, VIME), SHAP-PGD outperforms prior methods (CAPGD, CaFA, Sparse-PGD) in 35/40 settings, achieving stronger attacks with improved semantic realism and interpretability.

**Strengths:**

- In addition to satisfying constraints, this is the first work that attempts to preserve semantics consistency beyond Lp-norm perturbations in tabular adversarial attacks.
- In these settings, the combination of explainability (via SHAP) with adversarial optimization is original.
- Used datasets and models are relevant. It seems that Simonetto et al. have released a fifth one on Malware (https://arxiv.org/pdf/2408.07579), which could be interesting to verify the two hypotheses made in Section 3.2.
- Extensive experiments.

**Weaknesses:**

- Novelty is incremental. SHAP and gradient attacks are not new, and the only difference with previous work is their integration.
- The paper assimilates semantics consistency with preservation of statistical distribution and of model clean accuracy. While this makes sense to state such hypotheses, validating these hypotheses is important and necessitate dataset-specific analyses (automated or human-based).
- Unclear how much computational budget was given to each competing attack and whether the comparison is fair wrt this criterion.
- Complementarity between the attacks not study. It could make sense to check if the proposed method would be useful to integrate in Simonetto's CAA attack, which in my understanding combines CAPGD with MOEVA (referenced in the paper).

**Questions:**

- Why are Table 1 and Table 2 results on a different set of datasets?

- How did you align the computational budget across the different attacks?

- Section 4.3 is very descriptive; what conclusion we can draw from that? How does it compare to other attacks?

- Fig. 3, line 361: How can k be a real number and not a natural number?

- Line 367: How is the feasible success rate computed in practice? Is the attack restarted without constraints to compute the denominator? I am not sure if this metric is relevant since it seems to favor constraint satisfaction (no matter how many x are in X_adv, if all of them are constrained, the metric outputs 100%).

- Why do we have only 3 models in Table 6? I also did not find the rest in the appendix.

---

### Meta-Review · Area_Chair_X2jY · 2025-12-12

**Summary:**

All reviewers agree that the problem of adversarial attack on tabular data is with sufficient motivation. However, significant concerns were raised regarding the presentation of the submission. Some reviewers also raised concerns regarding the technical novelty and the issue of overclaim.

**Reviewer Concerns:**

The author did not provide rebuttal to address reviewers' comments.

**Reviewer Scores:**

The author did not provide rebuttal to address reviewers' comments.

---

### Decision · Program_Chairs · 2026-01-26

Reject